

# TMUB1 expression is associated with the prognosis of colon cancer and immune cell infiltration

Yan Lu[1,2,*], Kang Wang[3,4,*], Yuanhong Peng[3,4], Jun Zhang[3,4], Qinuo Ju[5], Qihuan Xu[3,4], Manzhao Ouyang[3,4] and Zhiwei He[1]

[1] Guangdong Medical University, Dongguan, Guangdong Province, China
[2] GCP Center, Shunde Hospital, Southern Medical University (The First People's Hospital of Shunde Foshan), Foshan, Guangdong Province, China
[3] Department of Gastrointestinal Surgery, Shunde Hospital, Southern Medical University (The First People's Hospital of Shunde Foshan), Foshan, Guangdong Province, China
[4] The Second School of Clinical Medicine, Southern Medical University, Guangzhou, Guangdong, China
[5] Guangdong Country Garden School, Foshan, Guangdong Province, China
[*] These authors contributed equally to this work.

Corresponding authors
Manzhao Ouyang,
doctorken@smu.edu.cn
Zhiwei He, hezhiwei@gdmu.edu.cn

## ABSTRACT

**Background**. *TMUB1* is a transmembrane protein involved in biological signaling and plays an important role in the stability and transcription of P53. However, its role in tumor remains unknown.

**Methods**. Using R language, the expression level of 33 cancer spectrum *TMUB1* was analyzed by the public database TCGA, GEO and HPA, the differential expressed gene (DEG) screening and protein interaction (PPI) network was constructed, and the differential genes of *TMUB1* in colon cancer were identified. The relevant signaling pathways were identified by gene functional annotation and enrichment analysis. The ssGSEA algorithm in GSVA were used for immune infiltration analysis. The Kaplan-Meier analysis, univariate and multivariate Cox regression analysis, nomogram and calibration map analysis were constructed to evaluate the correlation between *TMUB1* expression and clinical prognosis. The expression levels of *TMUB1* in intestinal cancer cell lines as well as in 10 intestinal cancer tissues were verified by qPCR experiments.

**Results**. Through the bioinformatics analysis of multiple databases and preliminary experimental studies, we found that the expression of *TMUB1* was significantly increased in colon cancer tumors, and was correlated with the clinical N stage, pathological grade, lymphatic metastasis and BMI of colon cancer. *TMUB1* may be involved in the regulation of the malignant progression of colon cancer. Meanwhile, patients with high expression of *TMUB1* mRNA had worse OS and DSS, and *TMUB1* expression was an independent prognostic factor for OS and DSS. It was further found that highly expressed *TMUB1* tissues showed low levels of immune infiltration and stromal infiltration.

**Conclusion**. We reported the expression level of *TMUB1* in colon cancer and analyzed its potential prognostic value in colon cancer through the bioinformatics analysis and preliminary experimental studies. The high expression of *TMUB1* is a negative prognostic factor for colon cancer patients. *TMUB1* may be a potential target for colon cancer.

## INTRODUCTION

Colon adenocarcinoma (COAD), the most prevalent histological subtype of colon cancer, primarily affects the intestinal mucosa and metastasises to nearby organs (*Wu et al., 2020*). Although patients with early colon cancer who underwent radical resection had a 5-year survival rate of more than 90%, most patients were diagnosed with late or metastasised cancer, resulting in a 5-year survival rate that was reduced to 10% (*Shah et al., 2016*). There has been a significant advancement in the clinical management of colon cancer, which is currently managed through surgery, radiation, chemotherapy, and targeted therapy. However, the patient prognosis is still poor due to late diagnosis, rapid development, and high transfer frequency (*Hao et al., 2020*). Therefore, exploring the molecular mechanism of colon cancer and determining new biomarkers for survival assessment and targeted therapy are priorities.

Transmembrane and ubiquitin-like (UBL) domain-containing 1 protein (*TMUB1*), also known as hepatocyte odd protein shuttling, is an encoding protein found in the proliferation process of the liver. *TMUB1* comprises 245 amino acids, including a nuclear export signal (NES) at the amino-terminal and a ubiquitin-like region (UBL; 121–175 aa) (*Chen et al., 2019*). *TMUB1* is a transmembrane protein, the N-terminal has a cross-membrane structural domain, and the C-terminal has two cross-membrane structures (*Della-Fazia et al., 2021*).

It has been reported that *TMUB1* can be shuttled between nucleus and cytoplasm and may be transmitted in this way by the nucleus' biological signals (*Castelli et al., 2020*). *TMUB1's* brain function and its cross-membrane ubiquitin protein are the subjects of most research in the central nervous system (*Della-Fazia et al., 2021*). Using the Cancer Genome Atlas (TCGA database), *Della-Fazia et al. (2021)* found increased messenger ribonucleic acid (mRNA) levels of *TMUB1* in 17 cancer groups and 21 cancers. It was also reported higher *TMUB1* levels in cancerous tissue (colorectal cancer, stomach cancer, and oesophageal cancer) compared with healthy tissues of the digestive system (*Della-Fazia et al., 2020*). However, research on the biological function of *TMUB1* is warranted, and its role in terms of the tumor and mechanism in colon cancer requires further study. In this study, we comprehensively evaluated the prognostic value of *TMUB1* expression in COAD patients from the Cancer Genome Atlas (TCGA) database. In addition, *TMUB1* expression in COAD was validated using data from GEO database and HPA database. In addition, we performed GSEA function and pathway analyses to further understand the biological mechanism of *TMUB1* in the pathogenesis of COAD. We also investigated the correlation between tumbler expression and gene alterations and methylation. In addition, the association of *TMUB1* in tumor microenvironment and immune cell infiltration was analyzed. The possible molecular mechanism of *TMUB1* interaction with tumorigenesis and tumor immunity was comprehensively analyzed and discussed.

## MATERIALS AND METHODS

### Acquisition of bioinformatics analysis data

The TCGA database project (https://portal.gdc.cancer.gov/) was used to download colon cancer (colon) identified by high-throughput sequencing RNA data (each 1 million base fragments (fragments per kilobase million (FPKM) format) and the corresponding clinical pathological information data. Herein, RNA sequencing data was converted from the FPKM format to the transcripts per million reads format. The study was conducted in accordance with TCGA recommendations, and informed consent was obtained before data collection. The Gene Expression Omnibus (GEO) database (http://www.ncbi.nlm.nih.gov/geo/) of the National Centre for Biotechnology Information was used in this study. A colon cancer sample group and a control group (healthy tissue, adjacent non-cancer tissue), with at least 10 samples in each group, were included in the study. Finally, four gene expression profiles (GSE83889, GSE9348, GSE23878, and GSE47756) were selected. R (version 3.6.3) was used for data normalisation and statistical analysis and visualisation. The HPA database (https://www.proteinatlas.org/) retrieved *TMUB1* genes for normal colorectal tissue and colorectal cancer tissue. The Institutional Review Board of Shunde Hospital Southern Medical University (The First People's Hospital of Shunde), approved this study with the approval number 20210730.

### Clinical significance of *TMUB1* expression in colon cancer

Receiver operating characteristic (ROC) analysis was used to compare *TMUB1* expression in colon cancer and the adjacent tissues, and the predictive value of *TMUB1* in colon cancer diagnosis was tested. The clinical prognostic information of patients with colon cancer included overall survival (OS) and disease-specific survival (DSS). The prognosis was analysed using Kaplan–Meier (K–M) analysis and univariate and multivariate Cox regression analysis. R package: Survival (version 3.2.10) was used for statistical analysis of survival data (*Liu et al., 2018*). R package (rms) was used to construct, analyse, and visualise lipograms and calibration diagrams.

### Integration of differentially expressed gene screening and construction of a protein–protein interaction (PPI) network

RNA sequencing data in level 3 HTSeq-Counts format for differentially expressed gene (DEG) was identified using R package DESeq2 (version 1.26.0) and ggploT2 (version 3.3.3) (*Love, Huber & Anders, 2014*). A protein–protein interaction (PPI) network was constructed by analysing the molecular interactions in the *TMUB1* protein list through the STRING database (https://cn.string-db.org/), and the relationship of the protein network was displayed (*Szklarczyk et al., 2019*). R package (igraph) was used for visualisation (*Mora & Donaldson, 2011*).

### *TMUB1* functional annotation and Kyoto Encyclopaedia of Genes and Genomes pathway analysis

Gene Ontology (GO) functional annotation and Kyoto Encyclopaedia of Genes and Genomes (KEGG) pathway enrichment analysis were performed for *TMUB1*. Statistical

analysis and visualisation were performed using version R3.6.3. Enrichment analysis was performed using clusterProfiler (version 3.14.3), and visualisation was performed using ggploT2 (*Yu et al., 2012*).

### *TMUB1* differential GSEA

Herein, R package's clusterProfiler was used to perform GSEA to clarify significant functional and pathway differences between the high *TMUB1* and low *TMUB1* groups. Data sets from MSigDB after the adjustment of $P < 0.05$ error detection rate, false discovery rate of $P < 0.25$, and enrichment of standardised scores ($|NES|$) > 1 were considered significantly enriched.

### Correlation analysis of *TMUB1* immune infiltration

Immune infiltration analysis of *TMUB1* was performed using the single-sample GSEA algorithm in R package, Gene Set Variation Analysis (*Bindea et al., 2013*). Twenty-four types of infiltrating immune cells were obtained. The correlation between *TMUB1* and the enrichment scores of the 24 types of immune cells was analysed using Spearman's correction method. The enrichment scores of *TMUB1* high and low expression groups were analysed using the Wilcoxon rank-sum test.

### Cell culture

Human intestinal epithelial cells (NCM460) were cultured in Roswell Park Memorial Institute 1640, comprising 10% foetal bovine serum. Human intestinal carcinoma cells (SW480, RKO, HCT116, LoVo, and HT29) were cultured in Dulbecco's modified Eagle medium with 10% foetal bovine serum and were incubated at 37 °C, under 5% $CO_2$ and 95% humidity. All cells were procured from CAS Cell Bank.

### Tissue specimens

Cancer tissue and adjacent tissue samples were collected from 10 cases. All clinical samples were stored in a liquid nitrogen tank after quick-freezing within 40 min in vitro. All tissue samples used for total RNA extraction had RNA protectants added to them. All tumor samples were approved by the ethics committee of our university, and informed consent was obtained from all patients and received written informed consent from participants of the study. A clinicopathological diagnosis of colorectal cancer was confirmed in the included specimens.

### RNA extraction

Total RNA was extracted from tissues using TRIzol reagent (TaKaRa, Tokyo, Japan) as per the manufacturer's instructions. PrimeScript RT reverse transcription kit (#RR047A, TaKaRa, Tokyo, Japan) was used to reverse transcribe the extracted RNA. The amount of isolated RNA was detected using the SYBR green polymerase chain reaction (PCR) kit (#RR820A, TaKaRa, Tokyo, Japan). Quantitative reverse transcription (qRT)-PCR was performed using the cfX-96 coupled real-time fluorescence quantitative system (No. 788BR07388 Bio-Rad, Hercules, CA, USA).

The primer sequences used were as follows:

5′-CTACCTCATGAAGATCCTCACCGA-3′(beta-actin, forward)

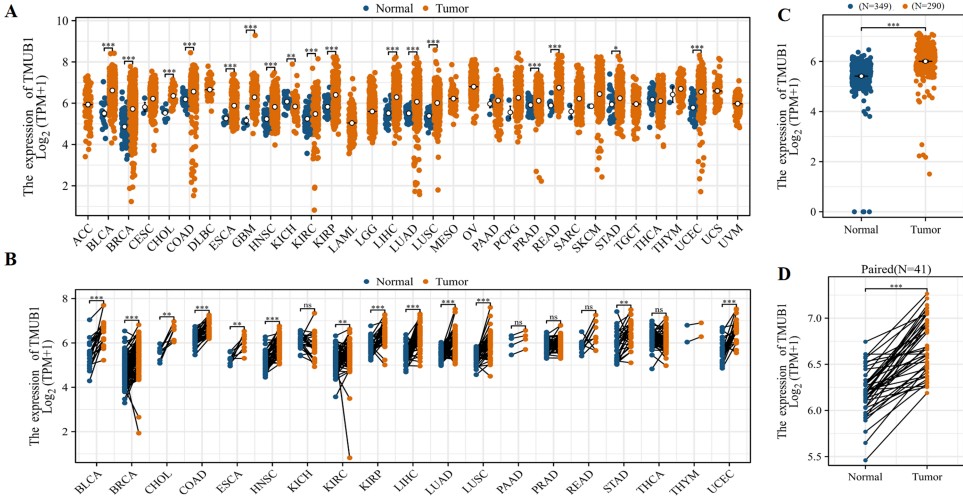

**Figure 1 Expression of *TMUB1* in colon cancer and other tumors.** (A) Expression of *TMUB1* in pan-carcinoma. (B) Expression of *TMUB1* in pancancerous paired samples. (C) Expression of *TMUB1* in COAD. (D) Expression of *TMUB1* in COAD paired samples. (*$P < 0.05$; ** $P < 0.01$; *** $P < 0.001$; ns $P > 0.05$, *vs* Normal).

5′-TTCTCCTTAATGTCACGCACGATT' (beta-actin, reverse)
5′-GTGTCCACGAGAGTCGGTC' (*TMUB1*, forward)
5′-AGGGCCGGTACTGGATCTG-3'(*TMUB1*, reverse)

## Statistical methods

Statistical analysis was performed using GraphPad Prism 8. The two sets of data were compared using an unpaired two-tail test. The chi-square test and rank-sum test were used to compare parameters. The correlation between variables was evaluated using Spearman's correlation analysis. The relationship between the *TMUB1* expression level and clinical features was analysed using univariate logistic regression. $P < 0.05$ was considered statistically significant (*, $P < 0.05$; **, $P < 0.01$; ***, $P < 0.001$).

## RESULTS

### *TMUB1* expression in colon cancer and other tumors

The TCGA database showed that *TMUB1* mRNA levels were highly expressed in different tumors. Among the 33 tumor types, *TMUB1* was significantly overexpressed in 23 tumors (Fig. 1A), particularly in gastrointestinal tumors (cholangiocarcinoma, colon adenocarcinoma, oesophageal carcinoma, liver hepatocellular carcinoma, rectum adenocarcinoma, and stomach adenocarcinoma). Similarly, *TMUB1* was found to be significantly overexpressed in gastrointestinal tumors in paired samples (Fig. 1B). Particularly, *TMUB1* expression was significantly higher in colon cancer tumors than in pericancerous tissues (Fig. 1C) and in paired tumors than in normal tissues ($N = 41$) (Fig. 1D).

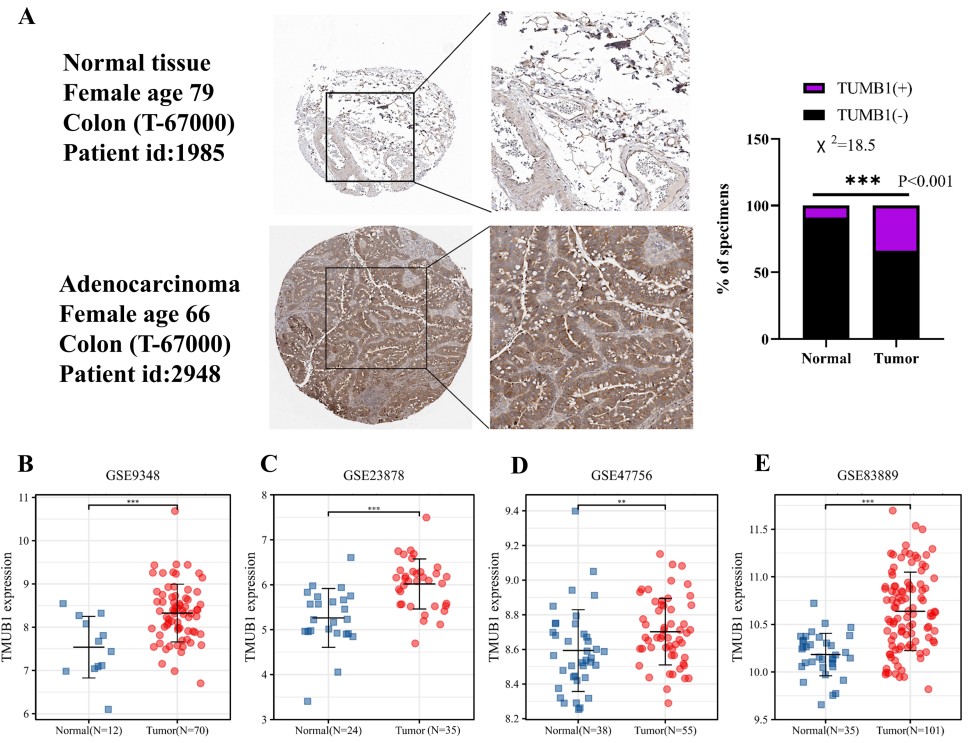

**Figure 2** **Expression of *TMUB1* in HPA database and GEO database.** (A) *TMUB1* protein was significantly overexpressed in colon cancer tissues according to HPA database (*** $P < 0.001$). (B) (E) Expression of *TMUB1* in GEO database. (B) GSE9348 (*** $P < 0.001$); (C) GSE23878 (*** $P < 0.001$); (D) GSE47756 (** $P < 0.01$); (E) GSE83889 (*** $P < 0.001$).

## *TMUB1* expression in the HPA and GEO databases

The HPA database revealed that *TMUB1* was significantly overexpressed in colon cancer tissues. In the HPA database, deeper immunohistochemical staining of cancer tissues compared with normal tissues was observed, suggesting that *TMUB1* was significantly overexpressed in colon cancer tissues (Fig. 2A). The GEO database data sets (GSE9348, GSE23878, GSE47756, and GSE83889) revealed that *TMUB1* expression in colon cancer was significantly higher than that in normal tissues (Figs. 2B–2E).

## Clinical characteristics of *TMUB1* in COAD and prediction of its diagnostic and prognostic value

We studied the clinicopathological features of *TMUB1* expression in COAD patients with differential topoisomerase II β-binding protein 1 interacting checkpoint and replication regulator expression, as shown in Table 1. The value of *TMUB1* in the differential diagnosis of COAD was proven using the ROC curve. An area under the curve (AUC) of 0.845 indicated that *TMUB1* was highly sensitive and specific in diagnosing colon cancer (Fig. 3A). K–M analysis verified the prediction of clinical survival using *TMUB1*, with a statistical significance in OS (Fig. 3B) (hazard ratio (HR) =1.98, $P = 0.002$) and DSS (Fig. 3C) (HR =2.23, $P = 0.003$). A poor prognosis was observed in colon cancer patients with a high *TMUB1* expression than in those with a low *TMUB1* expression. Table 2 presents

**Table 1  The clinicopathological features of *TMUB1* expression in COAD patients.** Differential topoisomerase II beta-binding protein 1 interaction checkpoint and replication regulator expression. The bold value is statistically significant ($P < 0.05$).

| Characteristic | Low expression of *TMUB1* | High expression of *TMUB1* | p |
|---|---|---|---|
| n | 239 | 239 | |
| Gender, n (%) | | | 0.647 |
|    Female | 116 (24.3%) | 110 (23%) | |
|    Male | 123 (25.7%) | 129 (27%) | |
| Age, n (%) | | | **0.020** |
|    ≤65 year | 110 (23%) | 84 (17.6%) | |
|    >65 year | 129 (27%) | 155 (32.4%) | |
| Weight, n (%) | | | 0.543 |
|    ≤90 kg | 117 (42.9%) | 72 (26.4%) | |
|    >90 kg | 48 (17.6%) | 36 (13.2%) | |
| T stage, n (%) | | | 0.168 |
|    T1 | 2 (0.4%) | 9 (1.9%) | |
|    T2 | 44 (9.2%) | 39 (8.2%) | |
|    T3 | 160 (33.5%) | 163 (34.2%) | |
|    T4 | 32 (6.7%) | 28 (5.9%) | |
| N stage, n (%) | | | 0.207 |
|    N0 | 151 (31.6%) | 133 (27.8%) | |
|    N1 | 51 (10.7%) | 57 (11.9%) | |
|    N2 | 37 (7.7%) | 49 (10.3%) | |
| M stage, n (%) | | | **0.012** |
|    M0 | 178 (42.9%) | 171 (41.2%) | |
|    M1 | 22 (5.3%) | 44 (10.6%) | |
| Pathologic stage, n (%) | | | **0.025** |
|    Stage I | 42 (9%) | 39 (8.4%) | |
|    Stage II | 103 (22.1%) | 84 (18%) | |
|    Stage III | 66 (14.1%) | 67 (14.3%) | |
|    Stage IV | 22 (4.7%) | 44 (9.4%) | |
| Perineural invasion, n (%) | | | 1.000 |
|    No | 84 (46.4%) | 51 (28.2%) | |
|    Yes | 29 (16%) | 17 (9.4%) | |
| Lymphatic invasion, n (%) | | | **0.001** |
|    No | 152 (35%) | 114 (26.3%) | |
|    Yes | 68 (15.7%) | 100 (23%) | |
| CEA level, n (%) | | | 0.921 |
|    ≤5 μg/L | 96 (31.7%) | 100 (33%) | |
|    >5 μg/L | 51 (16.8%) | 56 (18.5%) | |
| BMI, n (%) | | | 0.446 |
|    <25 kg/m$^2$ | 57 (22.3%) | 30 (11.7%) | |
|    ≥25 kg/m$^2$ | 101 (39.5%) | 68 (26.6%) | |

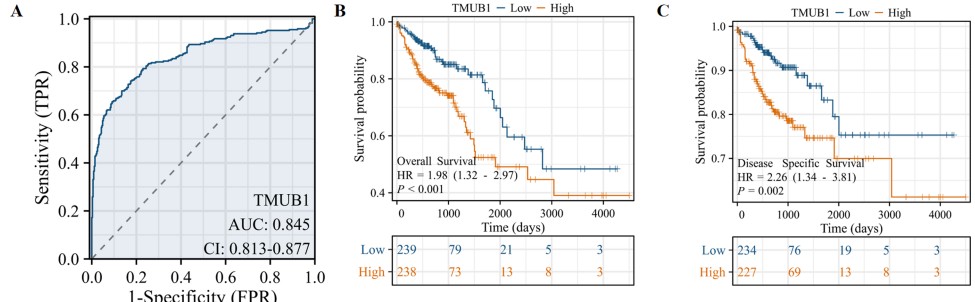

**Figure 3  Prediction of *TMUB1* in the diagnosis and prognosis of COAD.** (A) The value of *TMUB1* in the diagnosis of COAD was tested by ROC curve (AUC = 0.845). (B) Prediction of OS survival (HR = 1.98, *P* = 0.002). (C) Prediction of DSS survival (HR = 2.23, *P* = 0.003).

**Table 2  The correlation between *TMUB1* expression and the clinicopathological features analysed using logistic regression.** The bold value is statistically significant (*P* < 0.05).

| Characteristics | Total (N) | Odds ratio (OR) | *P* value |
|---|---|---|---|
| T stage (T3&T4 *vs*. T1&T2) | 477 | 0.953 (0.606–1.498) | 0.836 |
| N stage (N1&N2 *vs*. N0) | 478 | 1.368 (0.949–1.975) | 0.094 |
| M stage (M1 *vs*. M0) | 415 | 2.082 (1.210–3.675) | **0.009** |
| Pathologic stage (Stage III&Stage IV *vs*. Stage I&Stage II) | 467 | 1.487 (1.029–2.153) | **0.035** |
| Perineural invasion (YES *vs*. NO) | 181 | 0.966 (0.476–1.916) | 0.921 |
| Lymphatic invasion (YES *vs*. NO) | 434 | 1.961 (1.327–2.911) | **<0.001** |

the correlation between *TMUB1* expression and the clinicopathological features analysed using logistic regression. High *TMUB1* expression was associated with a high risk ratio in M-stage COAD (odds ratio (OR) = 2.082 [1.210–3.675], *P* = 0.009), clinicopathological advanced stage (OR = 1.487 [1.029–2.153], *P* = 0.035), and lymphatic metastasis.

## The significance of *TMUB1* in the clinical prognosis using univariate Cox regression analysis

The results of the univariate Cox regression analysis results based on the OS (Fig. 4A) and Table S1 showed that high *TMUB1* expression was associated with a higher risk of clinically adverse prognostic factors. *TMUB1* was a significant risk factor for clinical T3 and T4 stages (HR = 3.072, *P* = 0.004); N1 and N2 (HR = 2.592, *P* < 0.001); M1 (HR = 4.193, *P* < 0.001); Stages III and IV (HR = 2.947, *P* < 0.001); the presence of lymphatic invasion (HR = 2.450, *P* < 0.001). The prognosis based on the DSS has the same results (Fig. 4B). Further multivariate Cox regression analysis of the OS showed (Fig. 4C) revealed that high *TMUB1* expression was an independent risk factor for OS (HR = 1.917, *P* = 0.009). Similarly, high *TMUB1* expression was observed in clinical T3 and T4 stages (HR = 3.332, *P* = 0.047), M1 (HR = 1.906, *P* = 0.03), and stages II and IV (HR = 4.232, *P* = 0.021). Another independent risk factor was the presence of lymphatic invasion (HR = 1.771, *P* = 0.034). *TMUB1* had a significant predictive power for DSS (Fig. 4D, Table S2).

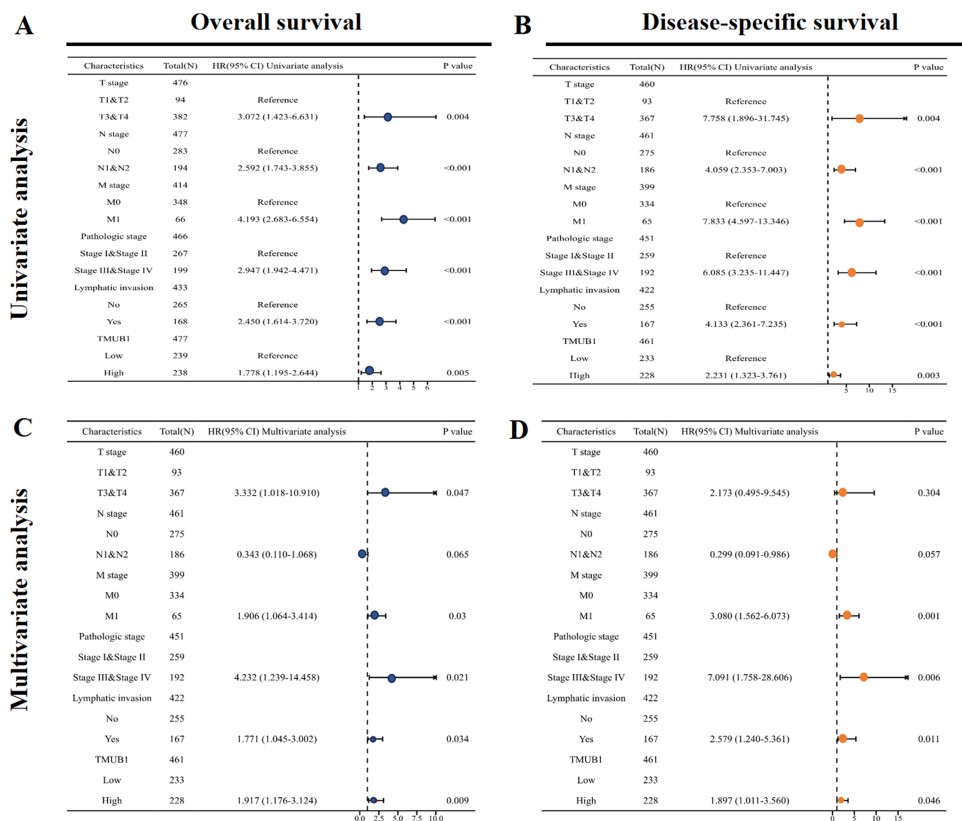

**Figure 4** **Significance of univariate and multivariate Cox regression analysis for clinical prognosis of *TMUB1*.** (A) Univariate Cox regression analysis based on OS. (B) Univariate Cox regression analysis based on DSS. (C) Multivariate Cox regression analysis based on OS. (D) Multivariate Cox regression analysis based on DSS.

## The predictive value of *TMUB1* in the diagnosis and prognosis of colon cancer

Clinical T stage, M stage, pathological stage, lymphatic metastasis, and *TMUB1* expression were constructed, and calibration curves were drawn to test the validity of the OS rolograms. The results revealed that the c-index of the OS rate of colon cancer was 0.747 (Fig. 5A). This helped predict the OS rate of colon cancer at 1 and 3 years in addition to predicting the prediction OS rate at 5 years (Figs. 5B, 5C, and 5D). Similarly, the M stage, the pathological stage, lymphatic metastasis, and *TMUB1* expression were used to construct the linear DSS prediction graph, and the c-index was 0.813 (Fig. 5E). Calibration curves revealed relatively accurate predictions of the clinical outcomes at 1, 3, and 5 years (Figs. 5F, 5G, and 5H). Similarly, in Fig. S1, DSS is also well predicted (Fig. S1).

## *TMUB1* in the clinical prognosis subgroup of colon cancer

The prognostic subgroups of *TMUB1* in terms of colon cancer OS and DSS were further evaluated. In the clinical subgroup K–M OS analysis, clinical T3 and T4 staging (HR = 2.35, *P* = 0.001), pathological stage (HR = 1.95, *P* = 0.010), age (HR = 2.08, *P* = 0.003), body mass index (BMI) (HR = 3.21, *P* = 0.011), and body weight (HR = 8.71, *P* = 0.041)

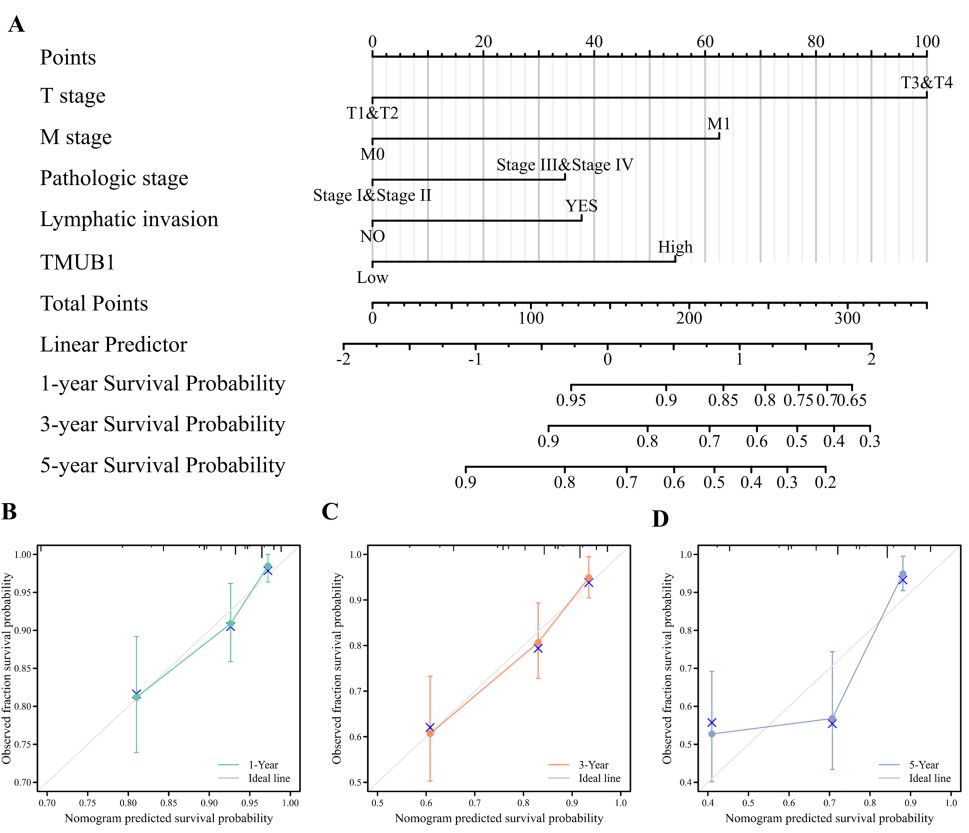

**Figure 5  The predictive value of *TMUB1* in the diagnosis and prognosis of colon cancer.** (A) Prognostic nomogram of *TMUB1* based on overall survival. (B–D) Calibration curves of overall survival at year 1, year 3, and year 5. (E) Prognostic nomogram of *TMUB1* based on OS. (F–H) Calibration curves of disease-specific survival at year 1, 3, and 5.

revealed a poor prognosis of *TMUB1* with high expression (Fig. 6A). Similarly, in the clinical subgroup K–M DSS analysis, clinical T3 and T4 staging (HR = 2.61, $P = 0.001$), N1 and N2 (HR = 2.06, $P = 0.018$), pathological staging (HR = 2.49, $P = 0.003$), and age (HR = 2.27, $P = 0.015$) revealed a prognosis of *TMUB1* with high expression (Fig. 6B).

## DEGs, co-expressed genes, and a PPI network of *TMUB1* in the TCGA in colon cancer

Variance analysis with volcanic figure illustrated $|log2\,(FC)| > 1$ and p.adj < 0.05. Number of molecules at the threshold of 0.05. Volcano shows meet $|log2\,(FC)| > 1$ & p.adj < 0.05. The 0.05 threshold had 1172 ids. Under this threshold, 128 ids had high expression (logFC positive) and 1,044 ids had low expression (logFC negative) (Fig. 7A). The top 10 genes positively and negatively associated with *TMUB1* are illustrated in the co-expression heat map (Fig. 7B). The top 20 interacting proteins associated with *TMUB1* by constructing PPI protein interaction networks are as follows: transmembrane protein (*TMEM*) 126A, autocrine motility factor receptor, family with sequence similarity 3 metabolism-regulating signalling molecule C, TMEM209, endoplasmic reticulum lipid raft-associated protein

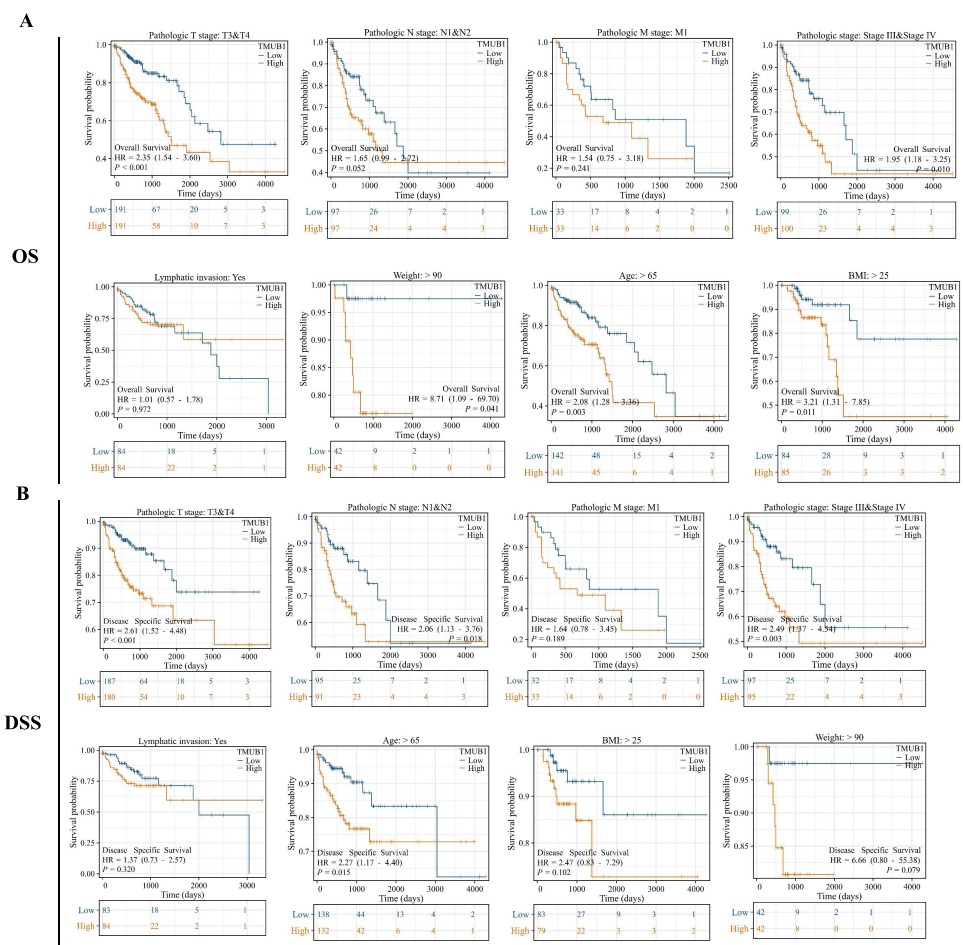

**Figure 6**  *TMUB1* prediction of clinical subgroup diagnosis and prognosis of colon cancer.  (A) Clinical subgroup K–M analysis of overall survival; (B) Clinical subgroup K–M analysis of disease-specific survival.

(*ERLIN*) 2, glutamate receptor interacting protein 1, saccharopine dehydrogenase, calcium-modulating ligand, *ERLIN1*, ring finger protein 139, *TMEM53*, protein interacting with C kinase-1, cleft lip and palate transmembrane protein 1, ubiquitin-associated domain-containing protein 2, fas associated factor family member 2, calsyntenin 3, E3 ubiquitin-protein ligase synoviolin, chromosome 17 opening reading frame 62, *NCLN*, and nurium (Fig. 7C).

## Functional enrichment analysis of *TMUB1*

By identifying differential genes, functional annotation analyses were performed, including bioprocess, cellular component, and molecular function of three types of GO and the KEGG pathway enrichment analyses. The findings of the GO analysis revealed that *TMUB1* was involved in ion transport, ion channel regulation, intercellular adhesion, and extracellular matrix composition (Fig. 8A). The KEGG analysis revealed that the neuroactive ligand–receptor interaction pathway, the calcium signalling pathway, the

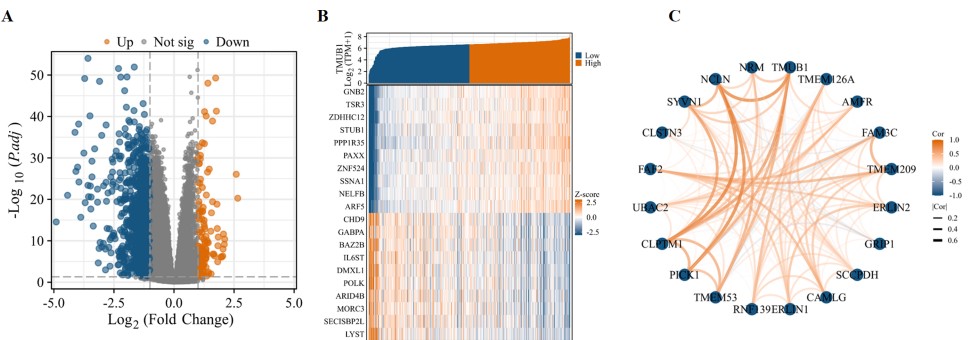

**Figure 7** **Differentially expressed genes and PPI network of *TMUB1* in colon cancer.**
(A) Differential genes for *TMUB1* in colon cancer, as shown in the volcano map. (B) *TMUB1* co-expressing gene; (C) *TMUB1*-related top 20 interacting proteins in the PPI network.

cyclic adenosine monophosphate signalling pathway, and the extracellular matrix (ECM)-receptor signalling pathways such as the interaction pathway were significantly enriched (Fig. 8B). The following *TMUB1*-related signalling pathways were identified by GSEA through comparative differential gene analysis: the insulin pathway, bladder cancer, glutathione metabolism, deoxyribonucleic acid (DNA) methylation, signalling by notch, and cell cycle checkpoints (Fig. 8C).

## Correlation analysis of *TMUB1* immune function in colon cancer

The degree of *TMUB1* immune infiltration in colon cancer was examined, and it was discovered that *TMUB1* was associated with a variety of immune cells. Figure 9A shows that *TMUB1* was positively correlated with CD56[bright] natural killer cells and other cells ($R = 0.268$, $P < 0.001$), and negatively correlated with T helper cells ($R = -0.407$, $P < 0.001$), central memory T (Tcm) cells ($R = -0.387$, $P < 0.001$), and macrophages ($R = -0.238$, $P < 0.001$). The immune infiltration and matrix scores were evaluated after *TMUB1* was divided into high-low expression groups, and the results revealed that the immune infiltration and matrix component scores of high-expression *TMUB1* were lower (Fig. 9B). Meanwhile, high-expression *TMUB1* in colon cancer showed lower infiltration levels in T helper cells, Tcm cells, T helper 2 (Th2) cells, and macrophages (Fig. 9C).

### *TMUB1* methylation level in colon cancer

We assessed the *TMUB1* methylation levels in colon cancer. The results revealed lower levels of methylation in colon cancer compared with normal tissues (Fig. 10A). Through Illumina methylation 450 (high-throughput methylation chip detection platform) and Spearman's correlation analysis, it was observed that in RNA sequencing data of colon cancer, *TMUB1* expression was negatively correlated with the beta corresponding to the methylation probe CG7847101 ($R = -0.329$, $P < 0.001$) (Fig. 10B). Among other features, lower *TMUB1* methylation levels in colon cancer were observed in TP53 mutations, adenocarcinoma categories, clinicopathological stage III/VI, and increased age (Figs. 10C–10F).

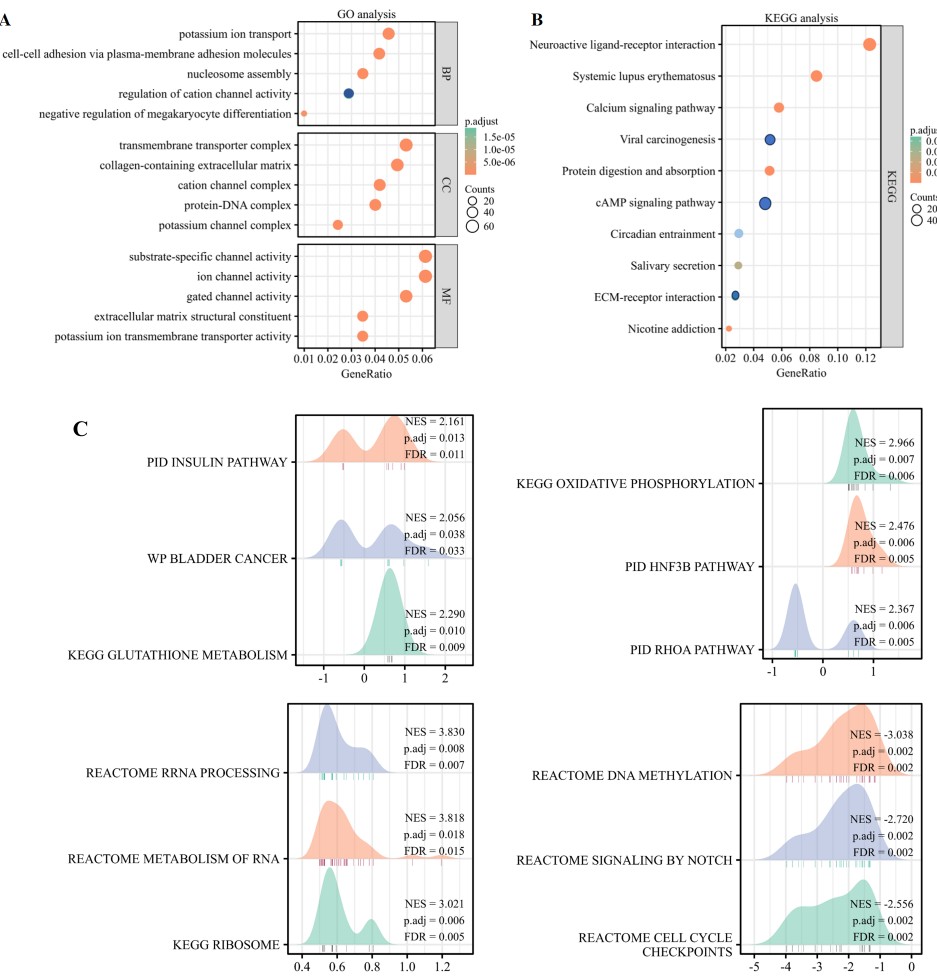

**Figure 8   Functional enrichment analysis of *TMUB1*.** (A) Gene Ontology analysis of *TMUB1* in colon cancer; (B) KEGG analysis of *TMUB1* in colon cancer; (C) *TMUB1*-related signaling pathway recognized by GSEA.

## qRT-PCR for verifying *TMUB1* expression levels in different colorectal cancer cell lines and tissues

To determine their expression levels in comparison to normal intestinal epithelial cells, human intestinal epithelial cells (NCM460) and human intestinal carcinoma cells (SW480, HCT116, RKO, LoVo, and HT29) were cultured. It was found that *TMUB1* expression in most intestinal carcinoma cells was higher than that in normal NCM460 cells (Fig. 11A). At the same time, paired colorectal cancer tissues from 10 patients were collected. Of the 10 paired colorectal cancer tissues (Fig. 11B), eight pairs had *TMUB1* expression levels higher than normal, while the remaining two pairs showed no difference (NATs: Normal Tissue Around tissues; tumor tissue).

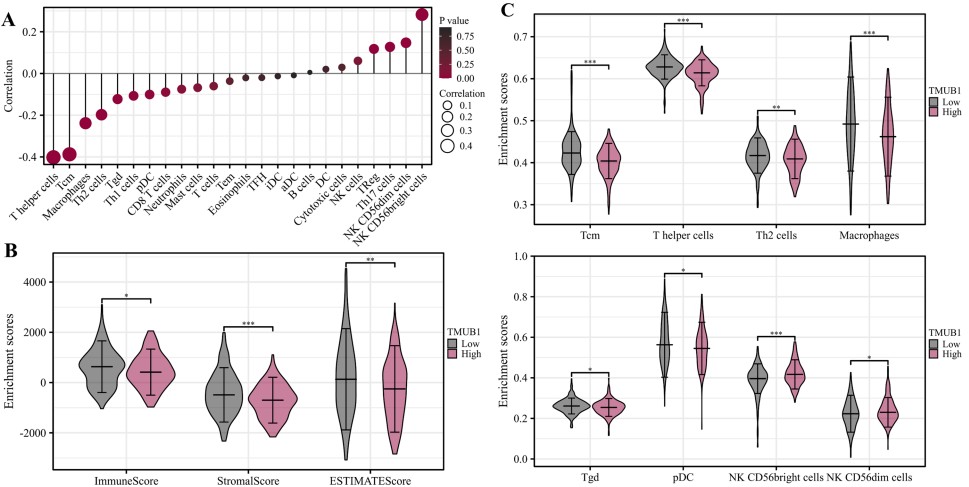

**Figure 9** **Correlation analysis of immune function of *TMUB1*.** (A) *TMUB1* and immune cell infiltration; (B) Scores of *TMUB1* and immune invasion and stromal invasion; (C) Comparison chart of different immune cell groups.

## DISCUSSION

COAD is a common malignant tumor. With current treatments, the therapeutic effect of COAD is far from satisfactory. Therefore, the search for stable potential biomarkers is essential to predict prognosis and guide individualized treatment. In this study, *TMUB1* was significantly overexpressed in most tumors in the TCGA data. Further analysis showed that *TMUB1* expression was significantly increased in colon cancer tumors. This result was also confirmed in GEO (colon cancer independent datasets GSE83889, GSE164191, *etc.*) and HPA data. ROC analysis of *TMUB1* expression may be a good diagnostic marker in colon cancer with an AUC of over 0.8. In further analysis of prognosis and clinical features, we found that *TMUB1* had a poor OS in COAD and was positively correlated with the stage of malignant progression. *TMUB1* is a promising prognostic marker for colon cancer based on its significant up-regulation in COAD, high diagnostic properties, and poor prognosis.

We compared various colorectal cancer cell lines with normal intestinal cells and detected their mRNA expression levels to verify the *TMUB1* expression level. Higher *TMUB1* expression levels were reported in human colorectal cancer (SW480, RKO, LoVo, and HT29) cells compared with that of the normal (NCM460) cells. No difference was observed in HCT116 cells. Based on the findings from 10 paired colon cancer tissues, it was reported that *TMUB1* expression in cancer tissues was higher than that in the adjacent tissues. This finding was consistent with the expression level in our TCGA and GEO databases, and further research to understand its mechanism in colon cancer is warranted.

Furthermore, researchers suggested that *TMUB1* is overexpressed in liver regeneration, shuttling between the nucleus and cytoplasm (*Chen et al., 2019*). Further analysis of function loss and function gain in human hepatocytes showed that *TMUB1* inhibited signal transducer and activator of transcription 3 (*STAT3*) phosphorylation and STAT3 signalling activation, thereby confirming the inhibitory effect of *TMUB1* overexpression

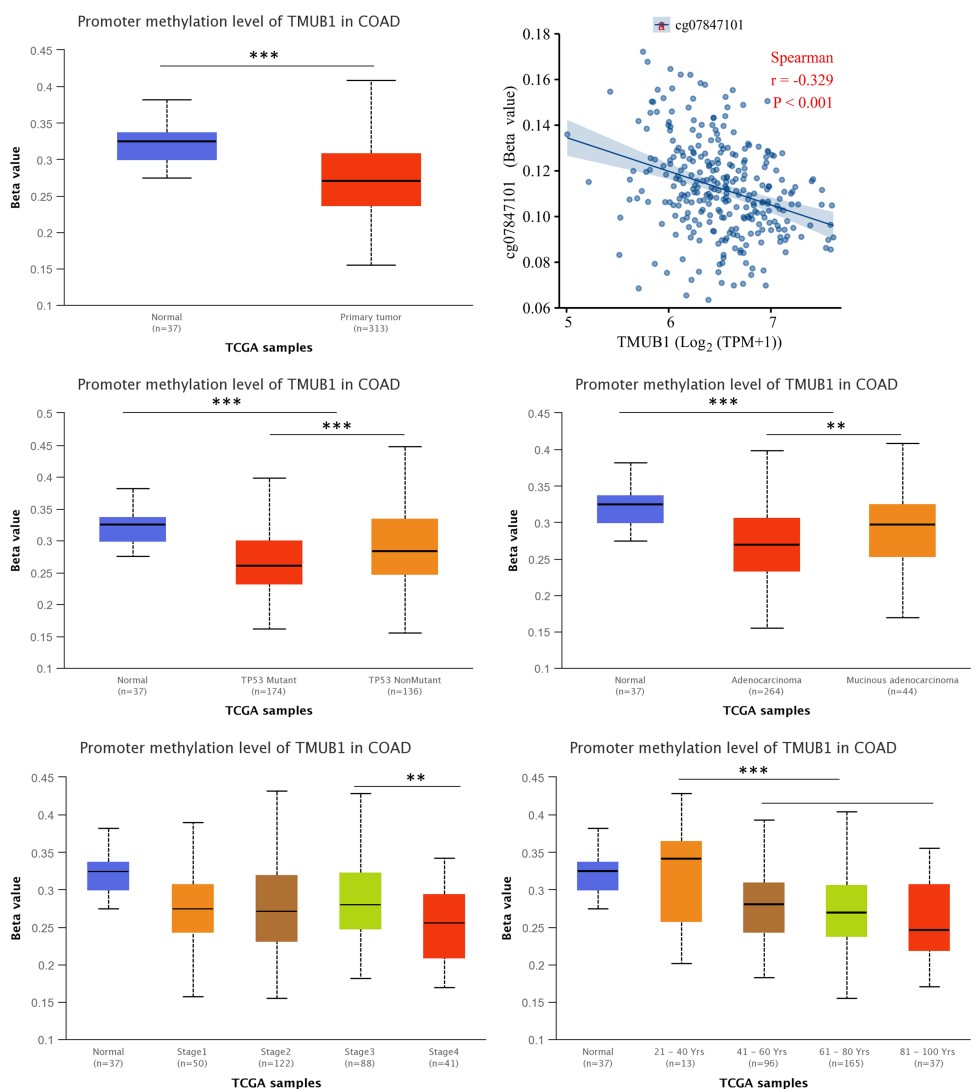

**Figure 10** **Methylation levels of *TMUB1* in different features of colon cancer.** (A) Methylation level of *TMUB1* in colon cancer (normal VS tumor); (B) The expression of *TMUB1* was correlated with the methylation probe CG7847101; (C) *TMUB1* methylation and TP53 mutation in colon cancer; (D) *TMUB1* methylation level and colon cancer tissue type; (E) *TMUB1* methylation level and pathological stage in colon cancer; (F) *TMUB1* methylation level and age group in colon cancer.

on hepatocyte proliferation (*Castelli et al., 2020*). *Hao et al. (2022)* founded that high expression of *TMUB1* was strongly associated with poor prognosis in colorectal cancer patients, which was consistent with our findings. Meanwhile, we focused on the expression profile, clinicopathological correlation and clinical significance of *TMUB1* by analyzing the TCGA-colon cancer dataset, and verified the expression level of *TMUB1* in colon cancer in cells and colon cancer tissues. Further we refined the validation that high expression of *TMUB1* was associated with the prognosis of colon cancer patients. It is suggested that high expression of *TMUB1* may be associated with poor prognosis of colon cancer patients. Other studies have revealed that *TMUB1* is crucial for the stability and transcription

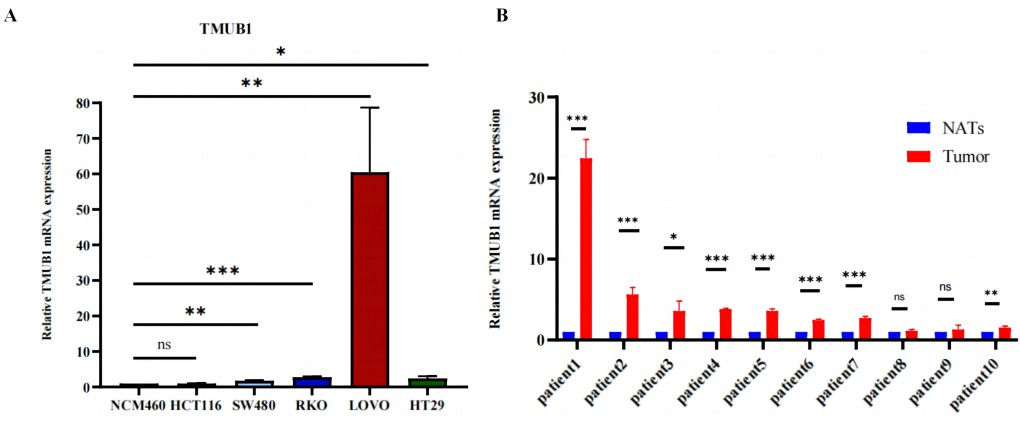

**Figure 11** **The expression level of *TMUB1* mRNA.** (A) The expression of *TMUB1* in SW480, HCT116, CACO2, RKO, LOVO and HT29 cells was higher than that in NCM460 cells. (B) The expression level of *TMUB1* in 8 of 10 paired colorectal cancer tissues was higher than that in normal tissues. (A) * $P < 0.05$; ** $P < 0.01$; *** $P < 0.001$; ns $P < 0.05$, *vs* NCM460; (B) * $P < 0.05$; ** $P < 0.01$; *** $P < 0.001$; ns $P < 0.05$, *vs* normal tissues.

of P53. It is found that *TMUB1* can not only increase P53 stability but also locate P53 in the mitochondria and promote mitochondrial apoptosis, indicating that *TMUB1* is crucial for controlling p53 stability and cytoplasmic accumulation (*Della-Fazia et al., 2020*). In addition, *TMUB1* was crucial for controlling important tumor suppressors such as nucleophosmin and auxin response factors (*Bellet et al., 2020*).

However, *TMUB1* was reported to be a poor risk factor for colon cancer in our study. Therefore, we predicted its possible function and signalling pathway in tumors by enrichment analysis to better understand its potential function in colon cancer. *TMUB1* was found to be significantly enriched in ion channel composition, cell adhesion, ECM composition, and ECM interaction in GO and the KEGG analysis. There have been many studies on tumor ECM remodelling, stromal cell infiltration, and ECM fibrogenesis causing tumor cell invasion and metastasis; however, several issues are yet to be resolved (*Wishart et al., 2020*). ECM fibrogenesis is expected to become a new type of anti-tumor metastasis target, but the mechanism of ECM fibrogenesis mediating tumor metastasis is still insufficient. For example, how tumor cells use ECM fibrogenesis to regulate the motility and adhesion of the tumor itself, and what are the differences between the arrangement and combination of different collagen molecules on tumor metastasis. Thus, *TMUB1* has the potential as a drug target for ECM development. The DNA methylation signalling pathway has gained attention in recent years among the multiple signalling pathways identified by GSEA (*Widschwendter et al., 2017*). Herein, we observed that *TMUB1* methylation level in colon cancer was lower in P53 mutation, adenocarcinoma category, clinicopathological stage III/VI, and increased age. Low-level methylation helps detect early precancerous lesions (*Witte, Plass & Gerhauser, 2014*), thereby indicating that *TMUB1* has the potential to be used as a biomarker for the early diagnosis and treatment of colon cancer.

Additionally, we analysed the potential relationship between *TMUB1* expression and immune cell infiltration. High-level *TMUB1* expression was associated with low-level immune and stromal infiltrations. *TMUB1* was reported to be negatively correlated with T helper cells, Th2 cells, Tcm cells, and macrophages. T helper cells are crucial in the pathogenesis of inflammatory bowel disease (IBD) and promote inflammatory responses in IBD and colitis-related colon cancer (*Pramanik et al., 2018*; *Ansaldo et al., 2019*). Th2 cells mainly secrete interleukin (IL)-4, IL-10, and IL-13 to promote humoral immunity (*Wibowo et al., 2021*). A reduced Th1/Th2 ratio has been reported to promote tumor growth, enhance tumor immune evasion function, and promote tumor metastasis and progression through abnormal secretion of related cytokines. Currently, the mechanism of the Th1/Th2 ratio imbalance in tumorigenesis and progression is still unclear, and further research is warranted (*Na et al., 2020*). Comparing Tcm cells to effector memory T cells and effector T cells, several studies have found that Tcm cells have superior persistence and antitumor immunity. Overall, we anticipate learning the molecular pathways that control memory T cell formation, which will be essential for developing rational approaches to optimise cancer immunotherapy (*Liu, Sun & Chen, 2020*).

Herein, we reported the expression level of *TMUB1* in colon cancer and analyzed its potential prognostic value in colon cancer through the bioinformatics analysis and preliminary experimental studies. *TMUB1* may have great potential as a tumor marker for diagnosing colon cancer. High *TMUB1* expression shows low immune cell infiltration, and we have made some predictions about the function of the gene itself and the specific signalling pathway; however, further research is required to confirm these results. In conclusion, our findings suggest that *TMUB1* overexpression is an independent adverse prognostic factor for COAD. P53 and DNA methylation signaling pathways may be the key signaling pathways regulated by *TMUB1* in OCAD. In addition, *TMUB1* mediates immune cell infiltration in the tumor microenvironment. This study shows that *TMUB1* as a prognostic biomarker for COAD highlights its potential as a predictive biomarker and immunotherapeutic target.

### Funding

This work was supported by the Natural Science Foundation of Guangdong Province, China (No. 2022A1515012315); the Discipline construction project of Guangdong Medical University (4SG22005G); the Beijing Science and Technology Medical Development Foundation (No. KC2021-JX-0186-94); the In-Depth Promotion of the Innovation-Driven Assistance Project in Foshan City (No. 2021043); the 2018 Foshan City Outstanding Young Medical Talent Training Project (No. 600009); and Foshan City's 14th Five-Year Key Specialty Project (no serial number). The funders had no role in study design, data collection and analysis, decision to publish, or preparation of the manuscript.

### Grant Disclosures

The following grant information was disclosed by the authors:

The Natural Science Foundation of Guangdong Province, China: No. 2022A1515012315.
The Discipline construction project of Guangdong Medical University: 4SG22005G.
The Beijing Science and Technology Medical Development Foundation: No. KC2021-JX-0186-94.
The In-Depth Promotion of the Innovation-Driven Assistance Project in Foshan City: No. 2021043.
The 2018 Foshan City Outstanding Young Medical Talent Training Project: No. 600009.
Foshan City's 14th Five-Year Key Specialty Project.

## Competing Interests

The authors declare there are no competing interests.

## Author Contributions

- Yan Lu conceived and designed the experiments, authored or reviewed drafts of the article, and approved the final draft.
- Kang Wang conceived and designed the experiments, performed the experiments, analyzed the data, prepared figures and/or tables, authored or reviewed drafts of the article, and approved the final draft.
- Yuanhong Peng performed the experiments, analyzed the data, prepared figures and/or tables, and approved the final draft.
- Jun Zhang performed the experiments, analyzed the data, prepared figures and/or tables, and approved the final draft.
- Qinuo Ju performed the experiments, prepared figures and/or tables, and approved the final draft.
- Qihuan Xu performed the experiments, prepared figures and/or tables, and approved the final draft.
- Manzhao Ouyang conceived and designed the experiments, authored or reviewed drafts of the article, and approved the final draft.
- Zhiwei He conceived and designed the experiments, authored or reviewed drafts of the article, and approved the final draft.

## Data Availability

The data is available at NCBI GEO: GSE83889, GSE9348, GSE23878, and GSE47756.

The TCGA database project (https://portal.gdc.cancer.gov/, search terms: colon adenocarcinoma, TMUB1, RNA-seq, and Clinical data set) .The HPA database (https://www.proteinatlas.org/, search terms: colon adenocarcinoma and TMUB1).

## Supplemental Information

Supplemental information for this article can be found online at http://dx.doi.org/10.7717/peerj.16334#supplemental-information.

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
