# Peer review of "TMUB1 expression is associated with the prognosis of colon cancer and immune cell infiltration"

_PeerJ, doi:10.7717/peerj.16334_

## Round 0.1 · original submission · Major Revisions

Please address the concerns of all reviewers and amend the manuscript accordingly.

Reviewer 1 ·

Basic reporting

1. The format is not unified throughout the manuscript. (ex. lines 92-106, no space between paragraphs 2.5 and 2.6 [line 139])
2. The language should be reviewed (ex. Line 159, no need to start a new sentence).
3. Use the italic font for the gene name.
4. The background provided on the TMUB1 structure is sufficient. No information on its function or pathway (please add).
5. The figures look great. Please add the abbreviations in the description (cancer names, TMUB1, COAD, etc..).
6. Overall, the tables look good and have sufficient data. Why are NO and YES in the tables capitalized? If they are not abbreviations, they should be written with the first letter capitalized only. Please add a table legend to explain the method used to analyze the data.

Experimental design

1. The article is original and within the journal’s aims and scope.
2. The research question needs more clarification. Why are you analyzing TMUB1 expression, clinical correlation, and survival prognosis? Are you looking for their effects on each other or on another aspect?
3. For the materials and methods section, I think that you wrote enough details, but the poor use of the language made it unclear.

Validity of the findings

1. More data should be written in the result section. It could be there, but the poor use of the language made it unclear.
2. I believe that the statistical methods used were correct.
3. The conclusions are not clear, and I couldn’t find a link to the research question as it wasn’t clear as well.

Additional comments

1. In the introduction, mention TMUB1 function and pathway.
2. In lines 74-76, you stated that “According to a study, TMUB1 can be shuttled between the body and the nucleus and may be transmitted in this way by the nucleus’ biological signals”. Please cite the study. What do you mean by the body? (Extracellular fluid, cell cytoplasm, etc.) Please be specific.
3. For lines 85-88, is this the proper place for those lines?
4. In section 3.1, you stated that “Among the 33 tumour types, TMUB1 was significantly overexpressed in 23 tumours,” what do you mean by overexpressed? What is the percentage? Additionally, you mentioned paired samples and tumours. Pair between what? (Genes, cancers).
5. In section 3.3, why you used the clinicopathological features of papillary renal cell carcinoma patients? And what is the relation between papillary renal cell carcinoma and COAD? Does Table 2 show the data of papillary renal cell carcinoma patients as well? (Please specify and mention it in the text and table title)
6. I believe that it could be a good article if it is rewritten properly. I suggest using manuscript proofreading and editing services and resubmitting then.

Reviewer 2 ·

Basic reporting

1, The English language and gramma should be improved to ensure that an international audience can clearly understand your text.
Line 159, line 433-434, subject like “ I” , “me”, “my” should be avoid.
Line 368 -370, a full stop is missing.
Line 414, “Thereby”, capital is not needed.
Line 74, 79, 393, “a study” “some studies” is not professional, the author should try to avoid those words.

2, The reference should be noted in correct position. Below are some examples:
Line 74, the author mentioned ”according to a study“, then there should be a refer at the end of this sentence. Ref 6 should be in line 76 instead of 77.
Line 370, ref14 is in wrong place.
Line 393, author mentioned "some studies" then there should be several related reference at the end of this sentence.
Line 407, " many studies" should also have corresponding references.

3, Other suggestions about writing:
Line 407 - 409, the author mention "several issues are yet to be resolved". The author should be clear about what are the issues, and how they are related to your current work. The author needs to first talk about previous study (please add reference about this), then talk about existing issue, then relate it to your work.
Line 372- 384, the author needs to rearrange the writing about findings/results from this study. Currently writing is diverted and wordy, hard to get the main point. The author shouldn’t just state the plain result from research, instead should use a logical discussion to connect each result and explain step by step, then lead to the final main point “TMUB1 can be a potential prognostic marker for colon cancer” . Similar issue in the following several paragraphs in the discussion section. The discussion part need to be improved extensively.

Experimental design

The main data and results from this work are from the database using analytic method, more experimental and wet lab methods are needed to provide strong evidence for the final conclusion. For example: a direct way to check the TMUB1 expression level in cancer cell lines or tissue, and make comparison to normal cell lines or tissue, this can be achieved by western blot or MS.

Validity of the findings

In another work from author "METTL27 is a prognostic biomarker of colon cancer and associated with immune invasion", the author conclusion that "The expression level of METTL27 was significantly higher in the colorectal cancer cell line than in normal cells, thus can be a potential prognostic marker of colon cancer". However, opposite result is observed in this work, the author mention that the methylation level in colon cancer is lower than normal cell in this study, and TMUB1 is related to methylation pathway. The author should have a discussion about this, since higher level of both METT27 and TMUB1 are observed in cancer cell.

Additional comments

N/A

Reviewer 3 ·

Basic reporting

Structure and Criteria: Seems appropriate.
Review the raw data: Whilst I am not completely familiar with the techniques used, the raw data seems appropriate.
Image check: The figures do not appear to have been manipulated.
Language: Writing in the work was largely good, however there were some areas that needed work…
Abbreviations: I would say there is often an overreliance on abbreviations. Often these abbreviations are not fully defined – especially in the figures. I would recommend including a list of abbreviations used at the end of every figure that relies heavily on them.
Sentence length: Please be careful with overly long sentences. They become very awkward to read. For example, In the Abstract the methods section has a sentence that is almost 100 words long. This is very difficult to read. As a rule, if a sentence is longer than 30 words you should think about rewriting for clarity.
Line 74 “TMUB1 can be shuttled between the body and the nucleus”: Do you mean the cell body?
Line 76 “TMUB1’s brain function and its cross-membrane”: Do you mean TMUB1 function within the brain?
Line 104 “cancer tissue.The”: Missing space.
Line 136 “TMUB1 groups Data sets”: Missing full stop.
Line 241 “HR=1,917”: Shouldn’t be a comma.
Line 254 “This helped predict the OS rate of colon cancer at 1 and 3 years in addition to predicting the prediction OS rate at 5 years”: This needs rewording. It doesn’t quite make sense.
Line 283 “Variance analysis with volcanic figure illustrated |log2(FC)| >1 and p.adj <0.05. Number of 284 molecules at the threshold of 0.05. Volcano shows meet |log2(FC)| >1 & p.adj <0.05. The 0.05 285 threshold had 1172 ids. Under this threshold, 128 ids had high expression (logFC positive) and 286 1044 ids had low expression (logFC negative)”: This may be due to my inexperience with volcanic figures, however I do not understand these sentences. What is “ids” referring to in this instance? What are your actual findings? Whilst I appreciate people familiar with the statistics might be more understanding, I do not believe that the majority of readers will be able to interpret this from what you have written.
Line 324 “<0.001) and T helper cells”: Do you mean “whereas” or something similar?
Line 325 “After TMUB1 was further divided into high-low expression groups, The immune infiltration and matrix scores were evaluated after TMUB1 was divided into high-low expression groups, and the results revealed that the immune infiltration and matrix component scores of high-expression TMUB1 were lower”: This needs rewording as it seems to repeat itself.
Line 341 “P & lt=0.001”: What does lt mean in this instance?
Line 369 “tumours Our”: Missing full stop.
Line 370 “Our study aims to focus”: Surely this should be past tense? You are now at the end of your study.
Line 394 “LO2 function”: What does LO2 function mean in this instance?
Line 400 “also locate P53 in the mitochondria and promote mitochondrial apoptosis, indicating that TMUB1 is crucial for controlling P53’s cytoplasmic function”: Surely this statement is referring to mitochondrial function rather than cytoplasmic functioning?
Figures and Tables: Well done and appropriate. There are some issues with clarity that I worry about – some of the text seems a little out of focus. This lets the interpretation down. Some further changes I would recommend.
Figure 1: A list of abbreviations is required to help interpret the figure. Furthermore, for 1C you have “normal” and “tumour” above and below the figure. Just below is adequate. 1D you have the n number above, this should also be done for 1C.
Figure 2: Rather than “(B) (E)” you should write “(B-E)”. For 2C, “Tumor(35)” should be changed to “Tumor(N=35)” for consistency. For B-E I also notice that the labels of the X axis are also above – this seems unnecessary as they say exactly the same thing.
Figure 3: TPR and FPR should be defined.
Figure 5: Rather than “(B, C, D)” and “(F, G, H)” you should write “(B-D)” and “(F-H)”. Also, there is no F-H on the figure – it seems to be missing. I cant seem to find any figure I missed – so I assume this was an oversight.
Figure 6: Panel A and B should be clearly labelled.
Figure 7: “TMUB1 in colon cancer The differential volcano map” needs to be reworded as it doesn’t quite make sense.
Figure 9: “Tcm” appears to be on the figure twice. Once at the beginning (next to Macrophages) and later on (next to Eosinophils). This is either an issue with results presentation or this is an issue with the text clarity. Either way, this is a problem.
Figure 10: Panels are not labelled (A-F). The X axes are illegible.
Figure 11: “TMUB1mRNA” is missing a space. There should also be a capital E at the beginning of the title (“Expression”). “NATs” should also be defined.
Table 1: You should include units for CEA and weight. It would also be good to include averages where relevant (age, weight, CEA, and BMI).
Table 2: Rather than do the n numbers for the total of the group, it should be separated based on the groups (i.e. M1 vs M0, rather than all the data for that group).

Experimental design

No comment.

Validity of the findings

All underlying data has been provided. Statistics seem appropriate. Figures have minor issues (see previous comments).

---

## Round 0.2 · Minor Revisions

Please address the remaining concerns of the reviewer and amend the manuscript accordingly.

Reviewer 2 ·

Basic reporting

The manuscript was significantly improved. But here are some issues that still need to be fixed.
1, Throughout the whole manuscript, the author sometimes uses TMUB1, and sometimes uses TUMB1. The author needs to correct the spelling. Also sometimes it is in italics, sometimes not. The author should check the whole manuscript carefully, and avoid such mistakes.

2, Line 318, "tumortumortumor" should be deleted

Experimental design

The author answered previous questions in response.

Validity of the findings

The author answered previous questions in response.

Additional comments

N/A

Reviewer 3 ·

Basic reporting

Good!

Experimental design

Good!

Validity of the findings

Good!

Additional comments

I would like to commend the authors on so completely fixing all my issues with their work. I think the work now reads much better, and the figures look excellent. After reviewing the tracked changes I can see how much effort has gone into addressing the reviewers comments, and I hope the authors feel the additional work has added to their publication.
I look forward to seeing this published.

---

## Round 0.3 · Minor Revisions

The authors state "We reported the expression level of TMUB1 in colon cancer for the first time, and analyzed its potential prognostic value in colon cancer." But, we found a reference reporting that TMUB1 is a marker for colon cancer.

Hao T, Yu H, Huang D, Liu Q. TMUB1 Correlated with Immune Infiltration Is a Prognostic Marker for Colorectal Cancer. Dis Markers. 2022 Jul 26;2022:5884531.

So it is not the first time, and this reference needs to be cited and the current results compared.

---

## Round 0.4 · accepted · Accept

Thank you for addressing the remaining issues and amending the manuscript accordingly. The revised manuscript is acceptable now.